# Power and interest levels in safely managed sanitation services in Zambia: A stakeholder mapping

Beatrice Chiwala[1]*, Mpundu Makasa[2], Joseph Mumba Zulu[1]

1 Department of Health Policy and Management, School of Public Health, University of Zambia, Lusaka, Zambia, 2 Department of Community and Family Medicine, School of Public Health, University of Zambia, Lusaka, Zambia

* bcchibwe2001@yahoo.com

## Abstract

### Background

Access to safely managed sanitation (SMS) in sub-Saharan Africa including Zambia remains a challenge. Variations in power and interest among stakeholder significantly influence access to SMS. However, there is limited contextualization of how power and interest levels among stakeholders influenced access to SMS. The study aimed to explore and analyze how stakeholders perceived their power and interest in the context of providing SMS. The study applied the Mendelow Stakeholder Matrix to identify, characterize and analyze the actors involved in the provision of SMS in peri-urban areas in Lusaka, Zambia.

### Methods

A narrative qualitative research design was employed in this study. Ninety–four (94) respondents participated in the study – 25 key informants who were representatives from Government Institutions/Departments, Cooperating Partners, NGOs and community level stakeholders; 60 discussants who participated in focus group discussions, while nine (9) community leaders took part in transect walks conducted in the target areas of Kanyama, Chawama and George. Nvivo14 was utilized for data management and analysis.

### Results

The main results suggested that stakeholders displayed interrelationships that were symbiotic as they depended on each other to deliver their mandates. Stakeholders categorized into Mendelow quadrants displayed varying levels of homogeneity in power and interest. In addition, some stakeholders such as the Local Authority shifted

**Editor:** D. Daniel, Gadjah Mada University Faculty of Medicine, Public Health, and Nursing: Universitas Gadjah Mada Fakultas Kedokteran Kesehatan Masyarakat dan Keperawatan, INDONESIA

**Data availability statement:** All relevant data are within the paper and its supporting information files.

**Funding:** "This research article is supported through the Norwegian Programme for Capacity Development in Higher Education and Research for Development (NORHED–II) and Strengthening Health Systems through Primary Care Leaders' Education (PRICE) project scholarship (Grant Number 70324) BC received in partnership with the University of Zambia. The funders had no role in the study design, data collection and analysis, decision to publish, or preparation of the manuscript.".

**Competing interests:** The sponsors and authors declare that they have no known competing financial interests or personal relationships that could have appeared to influence the study.

quadrants when seen to perform dual roles for example to implement and enforce the policies aimed at improved public health.

## Conclusion

The stakeholders' quadrant position coupled with persisted changes in their positions influenced their capacity to contribute effectively to the implementation of strategies to enhance access to SMS. This equally meant that implementers of SMS interventions have to regularly assess their engagement mechanisms to foster dialogue and coordination among stakeholders. Policy implications, especially to Government, may mean allocation of adequate resources to key players to enable them deliver on their respective mandates. Similarly, implications to practitioners might be the need to periodically review stakeholders and forge alliances coupled with conducting multi-sectoral meetings aimed to streamline their functions for the successful delivery of SMS.

## Introduction

Access to safely managed sanitation (SMS) remains an elusive goal for many worldwide [1]. Access to SMS entails improved sanitation facilities with safely managed human waste disposal, either in situ or off-site. [2,3]. A staggering 3.5 billion people across the globe in 2022 still lacked SMS [1]. Sub-Saharan Africa (SSA) alone contributed 24% of those without access to SMS which meant 2 in every 5 persons lacked these services [1]. In SSA, a majority of those affected lived in peri-urban and rural areas. They predominantly relied on non-sewered, unimproved sanitation facilities [1,4–7]. Provision of SMS to these settlements presented a complex phenomenon ranging from limited technological options compounded by high cost of implementing such technological options coupled with socio-cultural challenges [8–15]. Additionally, the high levels of urbanization and poverty made access to SMS an impossible goal to attain for most people in SSA [6,16–18]. The situation in Zambia is not any better as still 11 million people lacked basic sanitation [19,20].

Efforts have been made by various stakeholders ranging from Government agencies, Non-Governmental Organizations (NGOs), Community Based Organizations (CBO), Donors/Funding Agencies (DFA), and the Private Sector to try and improve access to SMS to non-sewered peri-urban and rural areas in SSA [6,15,21–23]. A stakeholder is defined as "any group, organization or individual that can influence or be influenced by the project" ([24], p319). The intricacy of ensuring all stakeholders worked together required robust stakeholder management strategies [25–28]. However, stakeholder management among sanitation actors has been a challenge due to among others: differences in priority areas; diversities in institutional mandates and implementation/institutional frameworks; available resources and capacities for stakeholder engagement [15,29].

Studies have suggested that stakeholder mapping was critical in the identification and management of stakeholders [30–32]. Differences in power and interest among stakeholder affect and influence access to SMS [33,34]. Therefore, stakeholder management is essential in developing strategies that address the stakeholders' needs,

concerns and perspectives in sanitation programs [35,36] Evidence has indicated that without stakeholder mapping, it would be difficult to know to what extent each stakeholder's power and interest influenced decisions of the organization or project [37–39]. Stakeholder mapping helps in defining and visualizing the relationships among stakeholders, enabling the creation of strategies to improve collaboration, meet expectations, and reduce potential conflicts [32,40,41].

Nevertheless, most studies in low and middle-income countries, Zambia inclusive, have focused on reporting challenges and possible solutions associated with sanitation. There is limited contextualization of how power and interest levels among stakeholders influenced access to SMS.

The objective of the study was to explore and analyze how stakeholders perceived their own power and interest in the context of providing SMS, utilizing Mendelow's stakeholder matrix lens to guide the mapping process [42]. The Mendelow's Stakeholder Matrix framework identifies and categorizes stakeholders into four quadrants based on their levels of 'power' and 'interest' in a project or organization [42,43]. The four quadrants are 'low power, low interest', 'low power, high interest', 'high power, low interest' and 'high power, high interest' [42,43]. The study further analyzed how the stakeholders' perceived quadrant position coupled with persisted changes in their positions had influenced their capacity to contribute effectively to implementation of strategies to enhance access to SMS.

## Methods

### Study site

We conducted the study in Lusaka city, the capital city of Zambia. We selected three peri-urban areas within Lusaka city namely Chawama, Kanyama and George to highlight and map key stakeholders who influenced the various sanitation interventions implemented in these areas over the years.

### The study design

A narrative qualitative research design was employed in this study. The narrative research design enabled the collection of data on stakeholders' roles in providing SMS at household level [44–47]. Using respondents' input, we identified stakeholders and evaluated their power interest levels through Mendelow's matrix.

### Mendelow stakeholder mapping matrix framework

The Mendelow's stakeholder mapping matrix framework was applied in this study to analyze stakeholder's attitudes and expectations along with their potential effects on the provision of SMS [1]. According to Mendelow, stakeholders are identified using two continuums namely "power" and "interest" [42,43]. He suggested that stakeholders with power had the ability to influence decisions in a project or organization. Similarly, Mendelow suggested that stakeholders could have diverse interests in a project or organization and outcomes. He argued that some stakeholders may have high power but low interest while others could have high interest but low power to influence the course of a project or organization [42,43]. Fig 1 denotes the Mendelow's stakeholder matrix model [42,43]. Walker et al suggested that stakeholder management was critical to the success of any project [32]. Therefore, stakeholder identification and mapping assisted in the analysis of the various actors' ability to influence and impact on the attainment of organizational objectives [48]. Mendelow argued that the attention one paid to the stakeholders depended on where the stakeholder was placed on the continuum [42]. Verhagen and Carrasco further suggested that "for sanitation service to work, interests of different stakeholders need to be well aligned, with the public sector playing their role"([49], p9). Therefore, conducting stakeholder analysis in the provision of SMS was essential for devising strategies aimed at enhancing engagement with various stakeholders across the continuum [49].

### Participants

We used purposive sampling to obtain a sample of 94 participants – 25 for key informant interviews, 60 for focus group discussion and nine [9] for transect walks. Each participant signed a written consent form to participate in the study.

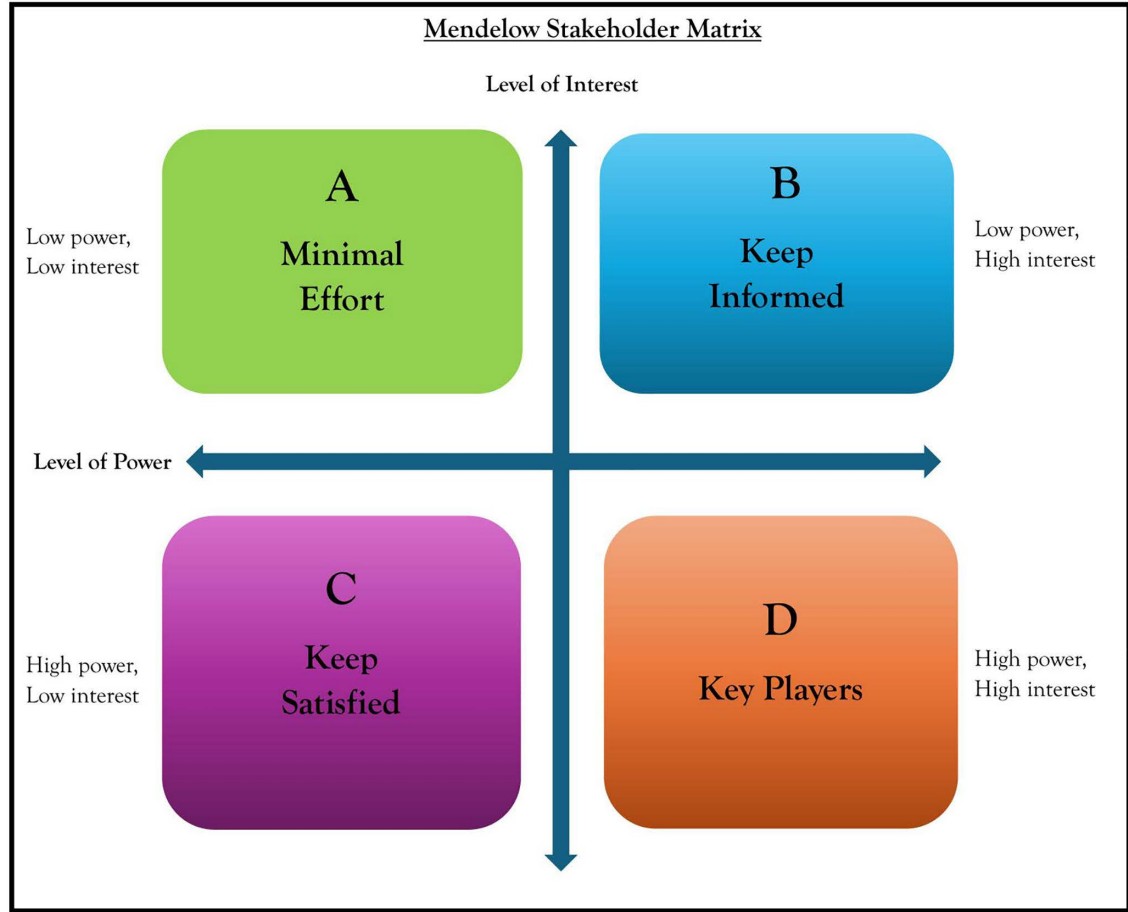

**Fig 1. Mendelow stakeholder matrix model.**

Key informants were recruited based on their respective roles among sanitation actors. Among them were representatives from Government Ministries/departments/agencies/institutions (including the national water and sanitation Regulator; relevant Ministries and the Utility responsible for water and sanitation in Lusaka); Cooperating Partners/Donors; Non-Governmental Organizations (NGOs) and selected community leaders from the target areas. Due to constraints in time and resources, not all stakeholders could be included in the study.

Transect walk participants (3 from each target area) were recruited from among the Water and Sanitation Committee Executive members. These committee members were those that participated in one way or another in the various sanitation interventions implemented in their respective communities. The Water and Sanitation Committees also assisted to recruit the focus group discussion participants both males and females who were heads of households with and without sanitation facilities at their homes. These community leaders assisted to recruit households because they were already familiar with the people they served, who had sanitation facilities or not.

## Data collection and sampling strategy

We collected both secondary and primary data in this study. The collection of secondary data began with the review of the Zambia legal, policy and institutional frameworks coupled with other relevant published literature written on sanitation and stakeholder management and engagements with majority on low- and middle-income countries. This provided insight into

the information available about sanitation and stakeholder management in low- and middle–income countries, particularly focusing on stakeholders involved in the provision of SMS in Zambia.

Primary data was collected through transect walks, focus group discussions and key informant interviews from 02nd August, 2022 to 25th September 2024. The study utilized transect walks as a method to observe sanitation practices. We utilized a specific observation tool during these walks, which involved both checking on the sanitation facilities and conducting brief interviews with randomly selected households. The walks were conducted early in the morning, from 06 AM to 07 AM for about 15 minutes, allowing the team to witness how households managed sanitation issues. Before each walk, the team planned what to observe, estimated duration, and determined the starting point. They opted for a zigzag transect walk pattern across all targeted areas to maximize observations within the allotted time. After completing the walks, the team convened for 15 minutes to share and discuss the findings. Both the observations during the walks and the discussions afterward were recorded for later analysis.

The study employed a focus group discussion guide featuring open–ended questions to gather data from heads of households. We conducted three separate focus group discussions in each area: one exclusively with 10 male participants, another with 10 female participants, followed by a combined session where both genders discussed together. Each session lasted 30 minutes and was audio–recorded for subsequent analysis. The discussions primarily took place in Bemba and Nyanja, local languages spoken in Lusaka city, and were later transcribed into English. The separation of groups aimed to explore gender dynamics related to sanitation. Focus group discussions also centered on exploring the extent to which households as key stakeholders understood and got engaged as beneficiaries in the provision of SMS.

We conducted a series of key informant interviews ranging from 30 to 80 minutes in duration. Our approach involved using a semi–structured interview guide consisting of open–ended questions to encourage detailed responses from participants. The interviews focused on understanding respondents' roles in sanitation service provision and their utilization of legal, policy, and institutional frameworks. We also explored their coordination efforts with other stakeholders and identified challenges they encountered. Additionally, respondents were asked about their influence over stakeholders' decisions, particularly concerning households' access to improved sanitation. Whenever possible, interviews were conducted face–to–face, with some sessions also held online using platforms like Zoom and Teams for the convenience of participants. Similarly, some interviews were conducted on phone. All the key informant interviews sessions were audio–recorded for subsequent analysis.

## Data analysis

All audio recorded transect walks, focus group discussions and key informant interviews were transcribed by an independent transcriber. BC sampled the transcripts against the audio files to validate their accuracy. All transcripts were cleaned and imported into NVivo release 14.23.2 [46] for coding and analysis. NVivo is a qualitative tool used for managing, organizing and analyzing data [50–52]. A reflective thematic analysis inspired by Braun and Clark [53] was applied to analyze all transcribed data using inductive and deductive coding approaches [54–57]. Inductive and deductive processes were applied during the coding process to enhance the rigor of created codes [55,56] Inductive approach was utilized to identify patterns on stakeholders' perceptions, views and ideas regarding their power and interest in the provision of SMS services. This process involved reading and re-reading the transcripts to identify significant information that would guide the mapping of stakeholders according to their power and interest. After identifying a series of codes, the research team explored connections between the identified codes and categorized stakeholders based on their similarities. Through the deductive process, the final codes were subsequently mapped onto the four quadrants of the Mendelow stakeholder matrix reflecting the sanitation actors' levels of power and interest. The subsequent stages involved reviewing and refining the themes to align with the study's objectives through a consultative process. Fig 2 show the final Mendelow stakeholder matrix highlighting the charaterization of sanitation actors based on their respective levels of power and interest in the provision of SMS in peri-urban areas of Zambia.

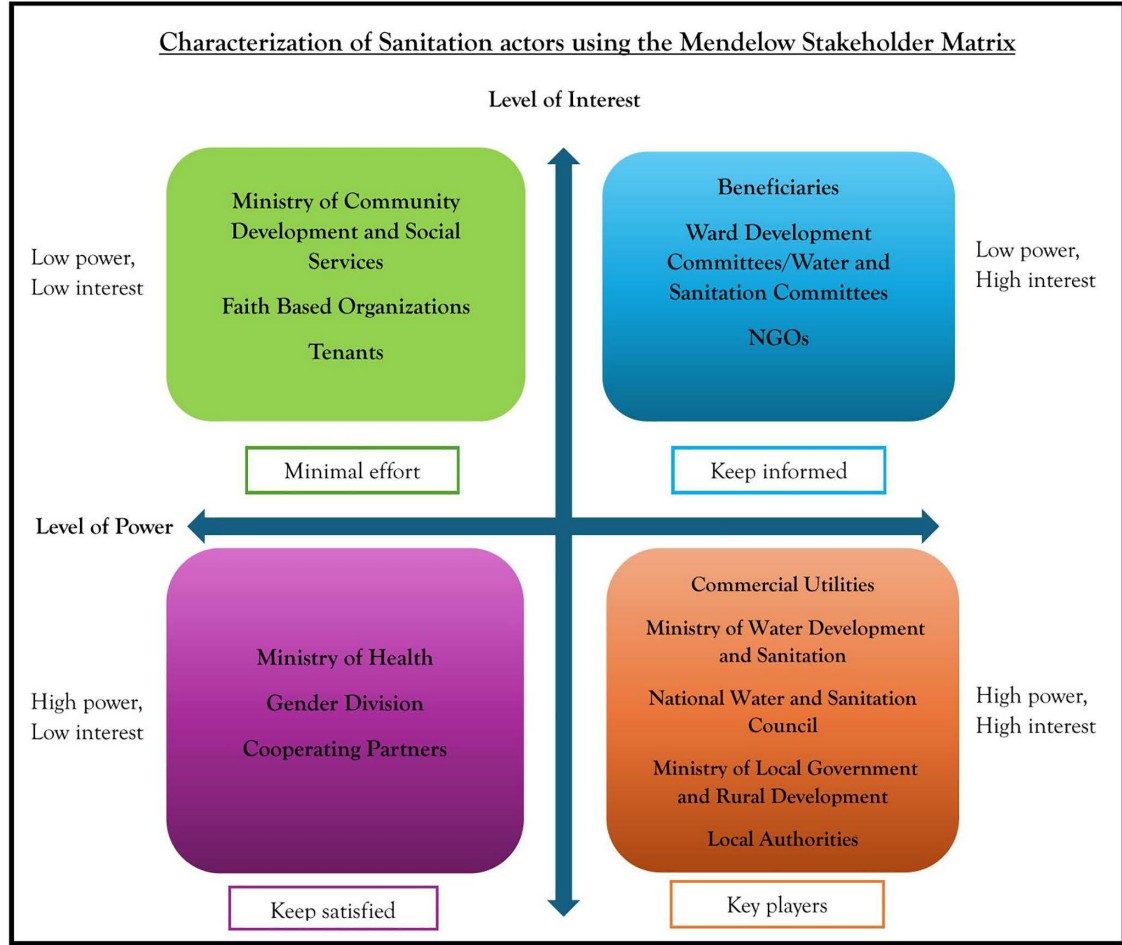

**Fig 2. Characterization of Sanitation actors using the Mendelow Stakeholder Matrix.**

The final stage of the analysis culminated in the production of this manuscript. Throughout the process, we engaged in consultations and discussions and took notes for future reference to keep to the study objectives.

## Ethics

Ethical approval for the study was obtained in November 2021 from Excellence in Research Ethics and Science (ERES), approval No.**2021–Sep–002**; and from National Health Research Authority, approval No: **NHRA00024/06/2022**. All participants for key informant interviews, focus group discussions and transect walks signed written consent forms. All respondents were anonymized and their responses kept in high confidentiality.

## Results

### Stakeholders and their roles

We began by categorizing stakeholders involved in SMS interventions to peri-urban areas based on their respective roles and responsibilities as shown in Table 1 below.

**Table 1. Initial stakeholders and their roles and responsibilities in SMS interventions.**

| Stakeholder | Roles and responsibilities in SMS interventions |
| --- | --- |
| National Water and Sanitation Council (NWASCO) | Government agency responsible for issuance of operating licenses, setting up minimum service standards for both on-site and off-site sanitation systems and monitors compliance among service providers to realize SMS. |
| Ministry of Local Government and Rural Development (MLGRD) | Government ministry responsible for orderly, coordinated and sustainable development of human settlements to facilitate access to SMS. |
| Ministry of Water Development and Sanitation | Government ministry responsible for policy formulation and mobilization of resources for SMS infrastructure development. |
| Local Authority (e.g., Lusaka City Council) | As principal shareholder in Commercial Utilities, local authorities are responsible for the provision of sanitation services, and enforcement of the Public Health Act. They also perform delegated responsibilities for other Government Ministries such as Ministry of LGHRD and Ministry of Health among others. |
| Commercial Utilities | Responsible for planning, implementation, provision and management of SMS interventions. |
| Cooperating Partners | Responsible for the provision of financial resources and technical expertise for sanitation infrastructure development to Government and service providers. |
| Ministry of Health | Government ministry responsible for making and enforcing policies related to public health. |
| NGOs | Responsible for advocacy, awareness raising and resource mobilization aimed at promoting SMS |
| Ministry of Community Development and Social Services | Government ministry responsible for various programs aimed at promoting equity, inclusion and advocacy for basic/minimum standard of living for the vulnerable mostly affected by poor access to SMS services. |
| Gender Division | Government division responsible for advocacy and awareness raising on matters of equity, inclusion and citizen participation. |
| Ward Development Committee (WDC) | Community-based organization responsible for enhancing community participation in planning, implementation and monitoring of SMS intervention initiatives. |
| Water and sanitation committee | As part of the WDC sub-committees, it is responsible for advocacy and awareness raising on SMS products and services besides promoting community participation in SMS initiatives |
| Faith Based Organizations | Plays a role in providing information aimed at promoting public health especially in times of emergencies. |
| Beneficiaries | Actual recipients and users of SMS interventions, and pay for these services. |
| Tenants | Occupants of houses or buildings rented out to them by property owners; they also fall in the category of beneficiaries of SMS facilities and infrastructure. |

## Mendelow's stakeholder mapping

Using respondents' input, we identified stakeholders and evaluated their power and interest levels through Mendelow's stakeholder mapping matrix. The process of setting up power and interest criteria was also informed by relevant literature. Tables 2 and 3 below are our adopted power and interest determinants in sanitation service provision respectively.

## Criteria to evaluate stakeholder power and interest levels

Informed by Mendelow's stakeholder matrix, we designed criteria to determine power and interest levels among stakeholders.

Four aspects determined power as shown in Table 2 below.

Table 2. Determinants of power in the provision of SMS.

| Power domain | Definition |
|---|---|
| Legislative or regulatory capacity | Possessing legislative or regulatory power to be able to create, enforce, or amend laws and regulations that could have an impact on sanitation projects [58–62]. |
| Financial capacity | Possessing control over resources– ability to release or withhold resources such as funding, material, personnel/expertise [59,62,63]. |
| Level of legitimacy | Normative perception of acceptance of each other's roles among stakeholders [59,64]. |
| Leadership position | Holding of strategic positions or roles that significantly influences the success or failure of sanitation projects [59–61]. |

Level of interested was assessed based on the following as shown in Table 3 below:

Table 3. Determinants of interest in the provision of SMS.

| Interest domain | Definition |
|---|---|
| Extent of impact | The extent to which a stakeholder is impacted or can influence decisions around sanitation. It also included the extent to which a stakeholder was aware of the positive or negative impact of SMS. |
| Willingness to participate | Desire of involvement or participation in sanitation projects. It also included the extent to which the stakeholder was engaged/consulted or aware of the sanitation interventions [65,66]. |
| Assessment of importance | Importance the stakeholder places on sanitation objectives, goals or outcomes [67,68]. |
| Adoption of opportunities | The extent to which a stakeholder can take advantage or not of opportunities to improve his/her sanitation status through the projects within his/her community [62]. |

## Characterization of actors using the Mendelow Stakeholder matrix

Utilizing the power and interest domains in Tables 2 and 3, we characterized the stakeholders' perceptions based on their roles and levels of involvement in SMS using the Mendelow's Stakeholder Mapping Matrix four quadrants as illustrated in Fig 2. We further grouped the stakeholders per quadrant as illustrated in Tables 4–7 below:

 

**Table 4. Low power, Low interest stakeholders.**

| Low power, Low interest |
| --- |
| Ministry of Community Development and Social Services |
| Faith Based Organizations |
| Tenants |

**Table 5. Low power, High interest stakeholders.**

| Low power, High interest |
| --- |
| Beneficiaries |
| Ward Development Committees/Water and Sanitation Committees |
| NGOs |

**Table 6. High power, Low interest stakeholders.**

| High power, Low interest |
| --- |
| Ministry of Health |
| Gender Division |
| Cooperating Partners |

**Table 7. High power, High interest stakeholders.**

| High power, High interest |
| --- |
| Commercial Utilities |
| Ministry of Water Development and Sanitation |
| National Water and Sanitation Council |
| Ministry of Local Government and Rural Development |
| Local Authorities |

**Low power, low interest**

In the first quadrant of **Low Power, Low Interest**, (Fig 2) we identified stakeholders as those who might not directly influence sanitation decisions. However, they still played an indirect role in creating an enabling environment that facilitated households' access to SMS. These were:

Ministry of Community Development and Social Services – The Ministry provided social protection and cash transfer services to vulnerable households.

*"We may not focus so much sanitation per say but then we tried to do awareness raising, we try teach beneficiaries the importance of how to use the money on sanitation aspect. So you find that the money that they get, they are able to use it to improve their houses, you know others are able to build houses from scratch, to you that money is very little it's just talk time and that but these households actually treasure it so much and are able to do quite a lot" (P15 KII).*

Faith-Based Organizations (FBOs) – FBO provided specific information to their congregants during public health crises, such as cholera outbreaks, to help prevent the spread of these diseases. However, they lacked influence on improved sanitation beyond the emergencies.

*"The church was very active in all the sectors that in the community, we are fully involved in the process, you know, and it was very easy for us to reach out to the people because you are speaking the same language, you are delivering the same message" (P24 KII).*

Tenants – Some tenants lacked the power to influence property owners to construct sanitation facilities. Similarly, tenants expressed reservations in investing in constructing sanitation facilities on rented properties.

*"Another thing that makes people not do have toilets in homes is the selfishness of some landlords because you find that tenants quite alright pay them but they do not use that money to build toilets and when the tenants are ignorant, they will not confront or encourage their landlords to build toilets" (TW1).*

**Low power, high interest**

In the second quadrant of **Low Power, High Interest** (Fig 2), we identified stakeholders as those who were very interested in sanitation outcomes but lacked the legal, policy, or institutional authority to provide or enforce these services. Additionally, some of these stakeholders lacked essential resources, such as finances, technical expertise, and materials, needed to invest in SMS. These were:

Non-Governmental Organizations and local Community Based Organizations such as the Ward Development Committees, and Water and Sanitation Committees – These stakeholders focused primarily on advocacy, raising awareness, and mediating between households and sanitation service providers.

*"We do not directly implement; we support the utilities and the government in their effort of providing better services to the community. But at the same time, we realize that communities also need attention. And this is why working through the utilities, we build the capacity, the capacity of communities to be able to understand one their commitment, but also the roles and responsibility for them to be able to have a viable or reliable sanitation service provision. In fact, one of our goals is a strong service provider" (P16 KII)*

Beneficiaries – These stakeholders were the actual recipients of sanitation services. Although they were affected by and highly interested in sanitation outcomes, they lacked the ability to directly influence project direction, resources, or priorities.

*"So the most important thing at every household is to have a toilet even when you are building a house you always make sure that there is a toilet and what has led some houses not to have toilets it's because of the heavy rainfall that we had so most of the toilets fell and others became full, otherwise even when you are looking for a house you always look for a toilet at any house" (FGD 5).*

**High power, low interest**

In the third quadrant of **High Power, Low Interest** (Fig 2), we identified stakeholders as those who, due to their legal mandates or financial positions, had the power to influence decisions but exhibited little interest in the daily operations or delivery of SMS services. Their primary concern was with deliverables, legal compliance, and adherence to service or financing agreements. These stakeholders had the authority to impose sanctions or penalties for failure to meet commitments by implementing agents, and played a significant role in ensuring public health for all citizens. These were:

Cooperating Partners – Cooperating Partners comprised of Donors/Financing Institutions who provided financial resources, such as loans and grants for sanitation interventions to Government or directly to implementing agencies like Commercial Utilities. Their financial strength gave them significant power to influence decisions, but they were less concerned with the daily operations or implementation of sanitation interventions. Their focus was on ensuring compliance

with financing guidelines and agreements. They also possessed technical expertise from their international exposure, which allowed them to offer insights into sanitation interventions from a global perspective.

*"our implementation framework basically is to support government initiatives, being the national development plans in the strategies which exists in the country in terms of rolling out sanitation, supporting the national programs for sanitation for water because that's the only way all these initiatives can trickle down to the communities or to the beneficiaries to improve access to sanitation" (P6, KII)*

The Gender Division – The Division had the legal mandate to promote and influence diversity, equity, and inclusiveness across all Government Ministries, departments, and institutions. However, results revealed that application, interpretation and enforcement of the Division's mandate at community and household level was limited as beneficiaries expressed concern that most toilets were not user friendly to the disabled, elderly, women and children.

*"So, our interest is to ensure that even the infrastructure that is to do with water takes care of the needs of women in terms of accessibility, usability, and also how maybe economically it could be, because if a certain services become too difficult for women to access, maybe because of the cost or because of distance where there available it becomes challenge for women. So, our interest is to ensure that these services are provided within reach and they must be affordable in terms of cost" (P11, KII).*

Ministry of Health – The ministry's mandate was to formulate public health policies, regulations and programs to ensure safe and hygienic living conditions for all. The ministry was also responsible for awareness raising and resource mobilization in collaboration with other stakeholders during public health emergencies to mitigate transmission risks. However, results showed that despite its considerable authority, the ministry's presence at community and household levels was limited to emergencies, which hindered its ability to influence household decisions regarding sanitation since this was not part of its core mandate.

*"Remember at the national level our role is to develop guidelines, protocols, standard operation procedures in consultation with other key line ministries………. We are looking at the ways how we can protect the public against the health hazards; here we are talking about prevention, suppression and control of diseases" (P14, KII).*

**High power, high interest**

In the fourth quadrant of **High Power, High Interest** (Fig 2), we identified stakeholders as those with the authority to formulate, shape and influence changes in laws, policies, regulations, and guidelines for SMS. In this quadrant are also those tasked with the daily operation and provision of sanitation services. Additionally, other stakeholders were those responsible for mobilizing resources such as finances, technical expertise, and materials to support sanitation interventions. Equally, some of these stakeholders had the power to enforce the laws. These were:

Ministry of Water Development and Sanitation (MWDS) – The MWDS was responsible for providing national policy direction in terms of formulating, amending and changing laws with regard to SMS. The Ministry was equally responsible for resource mobilizations such as finances, technical expertise and material to enable efficient and effective delivery of SMS services.

*"As Ministry of Water, we are more on the higher part where we look at resource mobilization and the policy direction to ensure that our line ministries or other organizations which are actually in the doing, are able to do those under a good environment" (P11, KII)*

National Water and Sanitation Council (NWASCO) – NWASCO was responsible for regulating the water and sanitation sector. The Institution had the authority to issue, cancel and enforce operating licenses to water and sanitation service providers (Commercial Utilities) as per license conditions. Additionally, NWASCO played an advisory role to Government and service providers on water and sanitation issues. Similarly, NWASCO periodically monitored and reported on the performance of service providers.

*"And in terms of the legal provision that we have that gives us the mandate to do that is, the Water Supply and Sanitation Act number 28 of 1997. So in that particular act, we have provisions that give us a mandate to regulate water and sanitation. And to zero in on sanitation, it has been defined as both on-site and off-site sanitation that we're dealing with and it has been streamlined to the removal of fecal waste" (P36, KII).*

Ministry of Local Government and Rural Development – The Ministry managed urban and regional planning to ensure the orderly development of human settlements. It delegated this core function to Local Authorities, who were equally responsible for providing essential municipal services such as water, sanitation, and drainage to make settlements livable.

*"Urban physical planning department is responsible for urban and regional planning in the country. The mandate is to ensure orderly, coordinated and sustainable development of human settlements. However, the ministry through local authorities also performs other functions that are delegated to the local authorities from other sectors such as public health. Now I'll actually zero in more on the role my department plays in relation to sanitation. Since our role is to ensure orderly, coordinated and sustainable, especially the part on sustainable, development of human settlements. We do plan for human settlements and control the development in the human settlements. In other words our role is to ensure people living in the settlements; they live in harmony and in a healthy and secure manner." (P39 KII)*

Local Authorities – Local Authorities, as sole shareholders in Commercial Utilities, superintended the provision of water and sanitation services under the direct supervision of the Ministry of Water Development and Sanitation. They also formulated and enforced by–laws to address public health nuisances. They were also conduits for executing various Ministries' mandates such as City development and planning under the Ministry of Local Government and Rural Development, and public health under the Ministry of Health.

*"So, we do enforcement like trying to visit all the areas and write the enforcement letters. To those people who do not want to comply by putting up toilet facilities within their premises or residentials areas otherwise it has been a challenge because planning has not, was not done in the planning in good faith we really missed it at planning stage. So where are we, are we very overwhelmed with problems that are there. So, we try to move in as much as it's not purely our mandate to provide sanitation as well as water supply services. So as our component is just to do enforcement and education, but obviously working hand with we other key stakeholders being the local authority" (P12 KII).*

Commercial Utilities – The Board Members, Managing Directors, Directors and employees on half of Commercial Utilities were responsible for provision and management of water and sanitation services on behalf of Government. By virtue of their positions, they possessed the authority to influence and make decisions that determined and ensured access to SMS. Commercial Utilities had the responsibility to adhere to service standards and commitments made to Regulators, customers, and Financiers. Using their technical expertise, Commercial Utilities also developed innovations in sanitation technologies to improve efficiency, reduce costs, and ensure sustainability. Additionally, they actively engaged with other stakeholders and made sure to include the voices of vulnerable groups in various sanitation initiatives.

*"Lusaka Water Supply and Sanitation Company basically has the mandate to provide this service. So it's within its jurisdiction to plan and budget for execution of a provisional sanitation at household level but the challenge is the same mobilizing the*

 

*resources because, most of the people who consider sanitation as not so much a profitable component of the service that we're providing and they're right because the tariff itself, for sanitation is very, very low. It does not cover even half of the operating costs required to operate sanitation facilities, the wastewater treatment plants, the collection systems, and so on." (P28 KII)*

**Stakeholder relationships in delivering safely managed sanitation mechanism**

Applying Mendelow's stakeholder matrix, the results showed how the stakeholders are interrelated in an effort to provide SMS. This inter–relationship arose because stakeholders in one way or another lacked the necessary resources, capacities, and structures to provide SMS effectively on their own. Below were the main interrelationships we identified:

Government Ministries/Regulator and Local Authority Stakeholders – Although they were responsible for formulating national sanitation policies and standards, some Government agencies such as the Ministries of Water Development and Sanitation; Local Government and Rural Development, and the Regulator NWASCO lacked the structures needed at community level to reach the intended beneficiaries. They therefore relied on Local Authorities and Commercial Utilities to execute these policies. This absence of direct community engagement caused inefficiencies, higher costs, and risks of policy misinterpretation. It also reduced accountability among agencies that were not part of the planning and development of sanitation interventions.

*"Yeah, maybe only the stakeholder challenges I have said we are moving for us to provide the service up to household level we will start as Ministry of Water then go to another Minister of Local Government then go to the person. So that one causes loss of time. Two it's expensive because you have to move resources from one ministry to the another. And accountability becomes a problem. Because you cannot hold an office and other Ministry directly responsible. You have to go through their permanent secretary and select" (P11, KII).*

Government and Donor/Cooperating Partner stakeholders – Despite having significant power, Government stakeholders lacked resources such as finances for capital investments in SMS. They collaborated with financial institutions and Donors who provided finances in form of loans and grants for sanitation infrastructure development.

*"We do not directly implement; we support the utilities and the government in their effort of providing better services to the community. But at the same time, we realize that communities also need attention. And this is why working through the utilities, we build the capacity, the capacity of communities to be able to understand one their commitment, but also the roles and responsibility for them to be able to have a viable or reliable sanitation service provision" (P16, KII).*

Commercial Utilities and community–level stakeholders – Due to limited capacity in community level engagements, Commercial Utilities collaborated with community–based organizations (for example the Ward Development Committees) and beneficiaries who despite wielded less power, played a crucial role in grassroots engagements and developing localized solutions.

*"So the decentralization policy, it shouldn't just be rhetoric, institutions or players involved in the whole process, I think they have to make sure that there is proper coordination in terms of the way they are implementing their policies, their programming and everything so that it reaches out to the grassroots because that's the whole purpose of having these policies to ensure that ordinary citizens benefit from the same" (P23 KII).*

**Stakeholders' attitudes and expectations**

Results also revealed that stakeholders displayed various attitudes and expectations of and towards each other. For example, the Regulator had expectations that the Commercial Utilities adhered to agreed service level guarantees and guidelines if their licenses were to be renewed. Similarly, the Donors who provided sanitation improvement loans or grants

expected the Government to abide by loan and grant agreements/conditions if they were to continue receiving the funds. Community level stakeholders also had their own attitudes and expectations towards sanitation service providers, for example, some households expected free sanitation facilities constructed for them.

*"What we look at is how are those initiatives aligned first of all to the national development plans of the country, in terms of ensuring improved access to sanitation in this case. And from there, also, we look at what alignment do these initiatives have in terms of the global, other Global initiatives like meeting the SDGs, and obviously meeting the aspirations of the Bank Water Strategy, in terms of where do you want to be in terms of Sanitation?" (P6 KII)*

Results also showed diverse expectations and aspirations among key stakeholders around the possibility of attaining universal access for all to SMS by 2030 for Zambia in general and Lusaka city in particular. Most of them alluded to it as an impossible task without increased investments in infrastructure development, human resource capacity building and robust stakeholder coordination mechanisms.

*"So the 2030 goal on sanitation it is not attainable. That is to start with the reasons that first of all, our investment in sanitation is very low. The rate at which we're going, the rate at which are investing is what is making me say we cannot achieve apart from the investment levels. The second is that sanitation one is sanitation has never been backed by adequate, legal and intuitional frameworks to push it forward, they're not adequate so we'll have to work in that area. Now what we need to do the is opposite of what I've just said. We need to ensure that we allocate more resources for sanitation. That is comprehensive sanitation, not only structures, but also the soft components capacity building for example"* (P11 KII)

### Shifting quadrants due to preforming dual roles

Results also showed that the position of various stakeholders' quadrants shifted from time to time depending on their roles at any given moment. We also observed that some stakeholders played dual roles as they shifted quadrants. For example:

Ministry of Health moved between quadrants of high power, low interest and high power, high interest when they stepped in to enforce the Public Health Act especially in times of public health emergencies like cholera outbreaks.

*"So once these protocols, guidelines, instructional frameworks are developed at the national level. They are channeled through provincial level, at provincial level the role is dual one is implementation and two is interpretation of the policies. When you look at the district and sub-district level is purely interpretation and implementation of these legal frameworks in order to reach out to the community" (P14, KII).*

Similarly, the results also suggested that community–level stakeholders shifted between low power, low interest and low power, high interest quadrants. For example, it was observed during the transect walks that some community members paid attention when one had sanitation challenges and could narrate such accordingly. In this instance, community members shifted from being disengaged to becoming more interested in sanitation matters, and felt a sense of responsibility to address their sanitation needs.

*"What makes people not to have toilets it's because of lack of money which makes it very difficult for them to construct toilets and also you talked about the law which talks about having a toilet at every household it's true but to have a standard toilet you require toilet with soak away whereby when they are full they can be emptied and most people did not know about those things it's just now that people have become aware about them." (TW1).*

The findings also revealed that dual roles created challenges, such as neglecting or downgrading functions not seen as central to the institutions' core mandates. Consequently, sensitization efforts were seen to be limited to emergencies. Each stakeholder's core mandate and interests influenced their level of participation in the provision of SMS.

*"There isn't sensitization going on in our compounds about sanitation which I feel is also a contributing factor. The public health is supposed to take lead in sensitizing people to ensure that they are aware about these issues because the challenge that we have is that when there is an outbreak of cholera that's when you see these people going round sensitizing people which becomes more costly as compared to sensitizing people even before things go bad and that will also help people to know and adhere" (FGD7).*

The dual roles could also be seen among some NGOs who performed the primary functions of advocacy and awareness also taking on the role of resource mobilization for sanitation infrastructure development in order to assist lower the cost of sanitation services at household level.

*"our financing mechanisms maybe you can put it like there are twofold so for the mandate service providers that we work with we look at it in terms of maybe helping them in terms of infrastructure development…… the biggest cost that is usually incurred is around the infrastructure." (P17 KII)*

Similarly, the Local Authorities faced an overload of duties, as they had to implement, interpret, and enforce mandates from other Government ministries and departments that lacked local level institutional structures.

*"So where are we, ah we are very overwhelmed with problems that are there. So, we try to move in as much as it's not purely our mandate to provide sanitation as well as water supply services. So as our component is just to do enforcement and education, but obviously working in hand with other key stakeholders being the local authority" (P12 KII).*

We also observed that some stakeholders had overlapping mandates in delivering SMS at the community level. For instance, the Ministry of Water Development and Sanitation, through Commercial Utilities, had an overlapping mandate with the Ministry of Health and the Ministry of Education in providing water and sanitation services to health facilities and schools respectively. Consequently, the Ministry of Water Development and Sanitation, which holds the primary mandate for these services at the national level, did not effectively track the progress made by the other two Ministries.

*"So, there are all these issues at institution or WASH as a mandate that remains with respective institutions like Ministry of Health, Ministry of Education, they also had institutional WASH mandate was reporting to who, who is tracking what is happening and this is why we are really unable to articulate the impacts that is being made towards achieving SDG six because everyone has something that they are doing and there's no central coordination on in terms of the data that is going". (P17, KII)*

## Limited collaboration and coordination among stakeholders

Results also revealed coordination challenges among stakeholders as some preferred to implement sanitation interventions in isolation. This contributed to duplication of efforts and negatively affected coordination mechanisms and resource mobilization in the sector.

*"I think the biggest challenge is working in silos. When we have a lot of partners, they always want to be visible, being seen as the ones spearheading everything, yes we may come, we may come together in the WASH committee"……..*

*But in terms of implementing, I think most of these institutions would want to be seen as the ones championing, improvement of sanitation in a specific area. So there's need to harmonize whatever we're doing and to avoid working in silos". (P33, KII)*

## Discussion

Our study analyzed the perceptions of key stakeholders in implementing SMS, utilizing Mendelow's stakeholder matrix lens to guide the mapping process [42,43]. Through application of this matrix, our findings suggested that the provision of SMS in peri-urban areas displayed a stakeholder relationship that was symbiotic. No one stakeholder possessed the monopoly of resources or capacity to provide SMS. This relationship was interdependent as each quadrant where various stakeholders belonged, contributed in various ways to facilitate achieving SMS. Our study also revealed significant differences in power and interest levels among stakeholders in each quadrant making their engagement and management more complex. Kapiriri and Razavi argued that stakeholders are not a homogenous group to be grouped in the respective four quadrants but that they are diverse and can fit in various categories [69]. For example, our study findings suggested that stakeholders in each quadrant performed different roles and could multi–task depending on circumstances. Stakeholders like the NGOs got involved in resource mobilization besides their primary function of advocacy and awareness raising.

We argue that implementation of SMS maybe an impossible task without stakeholders with power to influence and enforce decisions around sanitation. In similar studies, Chirgwin et al and Kennedy-Walker with her co–authors demonstrated that powerful stakeholders with policy, regulatory and financial power were strategic in ensuring there were resources to implement SMS besides creating an enabling environment in terms of legal, policy and institutional framework [8,62]. Other studies have also demonstrated that sanitation interventions especially in unplanned settlements was costly due to limited technological options coupled with social, cultural and high poverty levels [6,70–73]. Therefore, the powerful stakeholders due to their capacity to make decisions endeavor to find solutions to these issues. Their combination of power and interest can also accelerate progress in sanitation interventions [5,13,15]. However, studies have cautioned that the powerful stakeholder had the tendency to impose their decisions on the less powerful [64,74].

Stakeholder with low power, high interest who among them are beneficiaries or recipients/users are critical to sanitation interventions. Studies have suggested that without their buy–in and acceptability of the sanitation interventions, it would be very difficult for implementers to achieve their objectives [1,4,6,7,10,75]. This resonates with our study that suggests the importance of low power, high interest stakeholders in contributing to localized solutions and increased potential for sustainability of sanitation interventions. This has been echoed by Sutherland et al and Sutherland who argued that sanitation interventions needed to pass the social acceptability test from beneficiaries [11,76]. Studies have also shown that access to SMS is interwoven in gender and social inclusion issues which required localized solutions [25,77–80]. Mainstreaming gender in sanitation assisted in addressing unique gender needs for both women and men [1,77,81]. Similarly, mainstreaming social inclusion in sanitation assisted address issues of vulnerabilities such as access for persons with disabilities and the elderly [1,22,82].

Nevertheless, relying on low power, high interest stakeholders alone without those with high power, high interest might not be enough if SMS was to be attained. Studies have revealed that high poverty levels among the low power, high interest stakeholders, coupled with lack of land tenure for most residents in peri–urban areas compound the problem as revealed in this study and other studies [83–85]. As the results revealed, there were households that had no sanitation facilities for years despite the various interventions in the study target areas. This was where enforcement and regulation played a critical role to compel households to construct sanitation facilities [83,86]. Additionally as our results have shown, NGOs and community–level stakeholders also came in to raise awareness and advocate for improved sanitation at household and community level [87–89].

Our study has also confirmed that most sanitation interventions especially for peri-urban areas in Zambia had been implemented in an effort to abate public health emergencies such as cholera [90–94]. During emergencies, stakeholders with high power stepped in to provide resources, including financial, material, and technical support particularly for infrastructure development [93,95]. It was also during emergencies in our findings that stakeholders such as the Local Authority shifted quadrants and performed dual roles as they interpreted and enforced the Public Health Act. Stakeholder coordination to mobilize resources towards sanitation improvements also became easier in time of emergencies unlike other normal times [96–100].

Another important finding was the fact that stakeholders may not share the same vision and interests. They displayed differences in attitudes and expectations that could influence positively or negatively on sanitation interventions [101]. Peneno and Erickson suggested the need for a multi-stakeholder collaboration in order to create a shared vision [102]. However, the results revealed a variety of stakeholders with different degrees of power and interests who were all interested in one way or another in sanitation objectives, goals and outcomes and yet still held on to their respective mandates. This could be attributed to overlapping mandates coupled with gaps in the implementation framework. Kennedy-Walker et al and van Dijk have suggested that lack of clear policy and regulatory frameworks may result in overlapping mandates and duplication of effort challenges in the implementation of SMS [103–105].

On the other hand, findings revealed that some community members were unconcerned about the poor sanitation status around them. Such attitudes were usually associated with lack of information on the dangers of poor sanitation which may require a deliberate strategy to raise awareness on the importance of SMS targeting the community–level stakeholders [12,106–108]. Sanitation interventions subsequently required active engagement and participation of community– level actors if they were to be successful and sustainable [25,105].

## Alignment of the findings with mendelow stakeholder matrix and its influence on SMS outcomes

The findings in this study revealed a complex but fragmented stakeholder relationship in the provision of SMS to peri-urban areas of Lusaka city. Applying the Mendelow's stakeholder matrix, which contextually categorized stakeholders based on their power and interest levels, offers an analytical lens through which to interpret their behavior and formulate engagement mechanisms to realize SMS outcomes. This section will highlight how the stakeholder power- interest analysis and dynamic shifts in quadrants findings aligned or validated the Mendelow stakeholder matrix and their influence on SMS outcomes.

## Stakeholder power-interest analysis

The study findings suggested that stakeholders in the low power, low interest quadrant, among them tenants were passive participants in the provision of SMS. They also lacked the power to influence property owners to construct sanitation facilities despite paying them rentals. Studies have suggested that tenants in peri-urban areas were negatively affected by poor sanitation due to tenure insecurity and they lacked a sense of responsibility towards operation and maintenance of sanitation facilities [16,88,109–113]. Research has also indicated that tenants residing in these areas often accessed unhygienic sanitation facilities, typically shared by multiple households [83,114]. Mendelow's stakeholder framework suggested that such stakeholders should be engaged with minimal effort [42,43]. However, with increased urbanization of people from rural to urban areas in search of employment [68,115,116], implementers of SMS should engage tenants as one of the key players if SMS was to be realized in peri-urban areas. Tenants needed to be sensitized on their right to SMS and demand for it from their property owners. Through education campaigns, tenants could play a vital role in realizing improved sanitation outcomes in peri-urban areas.

In addition, our findings revealed that stakeholders with low power but high interest among them beneficiaries lacked essential resources to invest in SMS despite their high interest levels. Their high interest was critical to increased uptake,

sustainability and social acceptability of sanitation interventions at community and household levels [76,110,117]. Mendelow's framework recommended that such stakeholders should be kept informed and engaged [42,43]. Previous research has confirmed that sanitation projects where beneficiaries were actively involved yielded better results than those where they were not involved [118–121]. Without their active participation, it would be difficult to increase uptake and maintenance of SMS interventions coupled with its sustainability [6,122,123]. Though our findings affirmed this position, this cohort needed to be sensitized on the importance of investing in improved sanitation besides empowering them to demand for their rights from duty bearers like Government. Trimmer et al further suggested that the pro-poor subsidies coupled with construction guidelines may support vulnerable households to construct durable sanitation facilities [124].

Thirdly, stakeholders with high power but low interest such as financiers of sanitation interventions and Ministry of Health as our study has indicated played a critical role in influencing policy direction through their power to fund and enforce SMS respectively. Mendelow's model suggested that such stakeholders were to be kept satisfied by adherence to agreed terms and conditions [42,43]. Our study findings has affirmed that such stakeholders were more concerned about compliance to financing guidelines and agreements than day-to-day operations or implementation of SMS interventions. However, studies have revealed that over reliance on external support for sanitation infrastructure development in peri-urban areas sometimes undermined homegrown solutions to SMS due to limited participation of intended beneficiaries [9,124,125].

Lastly, stakeholders with high power, high interest such as Ministry of Water Development and Sanitation, Local Authorities, National Water and Sanitation Council and Commercial utilities emerged as critical actors in the provision of SMS. Engagement with these key players was essential in the delivery of SMS outcomes. Mendelow's model suggested that such stakeholders were to be closely managed [42,43]. However, our study findings suggested that these stakeholders displayed overlapping mandates and at times operated in silos due to fragmented institutional frameworks and limited operation/policy guidelines. Similarly, their lack of community level structures had the potential to derail the execution of their mandates directly to the intended beneficiaries. Studies have suggested that a lack of grassroots engagements often resulted into policies that were incoherent with aspirations of the intended beneficiaries [124,126]. Therefore, the study findings have underscored the importance of clarifying institutional roles and mandates. Additionally, these stakeholders required frequent inter-agency collaboration meetings to enhance shared goals and objectives. This would equally reduce on siloed interventions.

## Dynamic shifts

Provision of SMS in peri-urban areas revealed a complex stakeholder shift in power and interest due to changes in the operating environment. The study conducted by Kennedy-Walker in Zambia, suggested how sanitation was impacted by power, politics and history in Zambia [62]. Studies also have suggested that sector reforms in legal and policy frameworks coupled with the ever-evolving dynamics in peri-urban demographic and socio-economic due to urbanization caused stakeholders to shift their power and interest positions from time to time [62,105,124,127–129]. Similarly, a study conducted in Freetown, Sierra Leone, revealed that communities moved from being passive participants to demanding collaboration with the municipality authority [130]. Our study has also confirmed that stakeholders' quadrant position is not static, but dynamic in nature. Applying the Mendelow stakeholder matrix, may assist identify these dynamic shifts in stakeholders' quadrant positions and align engagement mechanism accordingly.

Additionally, the complexity in stakeholder dynamic shifts may result in sanitation actors performing dual roles and overlapping in mandates with other sector players. Studies have suggested that where roles were not clearly defined, conflicts of interests and misunderstandings among stakeholders are inevitable [131–136]. Applying the Mendelow stakeholder matrix in stakeholder engagements, may assist identify areas of overlaps and facilitate targeted conversations to ensure clear roles and responsibilities lines are drawn. The adaptability nature of the matrix may also assist sanitation implementers to quickly take remedial measures to realign their engagement strategies with the stakeholders' new position(s) in the quadrant. Some of the remedial measures may include legal, policy and institutional framework reforms to streamline the functions of stakeholders.

Similarly, the Mendelow matrix may assist sanitation implementers monitor and track stakeholders' movements overtime. The non-static nature of stakeholder positions if not monitored may result into sanitation implementers to be caught unaware when a major shift happen in the quadrant [133,134,137–140]. Studies have suggested that stakeholders such as tenants had played a major role in the implementation and management of sanitation in peri-urban areas [141–145]. Our study findings have also alluded to the fact that tenants with increased awareness on their rights to sanitation can shift quadrants from low power, low interest, to low power, high interest thereby becoming active participants in the delivery of SMS. Therefore, applying the Mendelow matrix may assist sanitation implementers keep track of stakeholder quadrant movements for the purpose of informing engagement strategies, planning and resource allocation to realize SMS outcomes.

Lastly, applying the Mendelow stakeholder matrix may assist to identify areas requiring improvements in guiding engagement strategies based on the prevailing stakeholder quadrant. Our study identified Local Authorities and Ministry of Local Government and Rural Development as key players in city planning and development. However, our findings highlight uncoordinated urban development planning and incomplete institutional frameworks especially at community-level resulting in overlapping mandates and inefficiencies in the delivery of SMS. The continuous growth of unplanned settlements posed a major challenge to provision of SMS to these areas [127,146–148]. Our study has suggested that these stakeholders despite having high power failed to enforce it to stop the growth of unplanned settlements. The Mendelow matrix lens may assist sanitation implementers to identify areas of weakness and devise empowerment mechanisms especially in the area of enforcement in order for them to manage the expansion of unplanned settlements and thereby enhance access to SMS. Similarly, identifying areas of strengths may assist sanitation implementers to leverage on the opportunities to invest in SMS.

### Strengths and limitations of the study

Though the Mendelow's stakeholder matrix offered valuable insights into stakeholder identification, categorization and management, it is important to acknowledge that the matrix has some limitations. The ever–changing dynamics in stakeholders' behavior, perceptions, priorities and interests made permanent categorization almost an impossible task. Similarly, stakeholders who shifted their quadrants from time to time to perform none primary roles contributed to the complexity of stakeholder management.

The other limitation laid in the complexity surrounding provision of sanitation to peri-urban areas that went beyond stakeholder categorization and engagement mechanisms. The Mendelow's stakeholder model present challenges in considering other factors that interface with provision of SMS to peri-urban areas. Studies have suggested other factors such as high poverty levels, unplanned nature of peri-urban areas, high urbanization and climate change as barriers to SMS in peri-urban areas [92,149–151]. Therefore, implementation of SMS to peri-urban areas needed to take into account these challenges associated with unplanned settlements besides stakeholder analysis if engagement strategies were to be effective.

Additionally, the fact that stakeholders were not a homogenous group, implementers needed to continuously forge alliances to keep them engaged and focused on improved sanitation outcomes [152].

Lastly, due to limited resources and time, the study was focused on a small sample whose perceptions though valuable, might not have been representative of all stakeholders. However, the findings have enabled us gather rich insights from sanitation stakeholders and thereby provide a contextual understanding into their varying power and interest levels. The study has also suggested how sanitation implementers can leverage on these dynamics to enhance delivery of SMS to peri-urban areas of Lusaka and potentially other similar contexts.

### Conclusion

The objective of the study was to explore how stakeholders perceived their own power and interest in the context of improved sanitation services, utilizing Mendelow's stakeholder matrix lens to guide the mapping process. The study

further analyzed how the stakeholders' perceived quadrant position coupled with persisted changes in their positions had influenced their capacity to contribute effectively to implementation of strategies to enhance access to SMS.

The main results suggest that stakeholders displayed an interrelationship that were mutual as they depended on each other to deliver their mandates. Similarly, stakeholders within and outside the quadrant were not homogenous. They each had varying levels of power and interest thus limiting the Mendelow's stakeholder matrix in identifying and analyzing all the sanitation actors. The other critical finding was that some stakeholders shifted quadrants when seen to perform dual roles – for example to implement and enforce the public health related policies in the case of the Local Authority.

The policy implications based on the findings are that stakeholder management was a continuous process that implementers of SMS needed to review in an effort to employ strategies that will resolve any interrelationships challenges that may arise. Achieving SMS required collective efforts from all stakeholders, regardless of their of power and interest levels. Policy implications at Government level are to prioritize stakeholders especially those with high power, high interest to ensure they were supported with financial, material and technical resources and accompanying legal and policy reforms to deliver on their respective mandates. Similarly, Government should ensure mechanisms were in place to make community/beneficiary participation mandatory for all sanitation projects and programs. This will enhance homegrown solutions and sustainability of sanitation interventions.

Practical implications to stakeholders with high power, high interest as key players is to enhance multi-sectoral collaborative platforms/meetings where they could discuss and forge a united front to tackle the numerous sanitation challenges in peri-urban areas. Targeted efforts such as signing memorandum of understanding (MOUs) with NGOs and other community-level stakeholders to be encouraged to assist reach to intended beneficiaries where institutional structures were lacking. Conversely, stakeholders in low power, high interest quadrant, notably beneficiaries (tenants inclusive), required empowerment with information on the importance of SMS to be able to participate meaningfully in the planning and management of SMS strategies.

This study has therefore, underscored the importance of contextual stakeholder analysis and management before implementing any sanitation intervention to enable effective coordination among sanitation actors. The study has also suggested that no one stakeholder can work alone to enable attainment of SMS. It required a consented effort from all actors with different capabilities and abilities, all working together in a symbiotic relationship to attain sanitation outcomes. However, sanitation project implementers should be mindful that no stakeholder level of power and interest is permanent. This change may emanate from the changing views, perceptions, opinions, and positions of stakeholders, along with shifts in operating environments. Hence the need for continuous revision of stakeholder management and engagement strategies.

Based on the findings, our recommendation to policy makers is that stakeholder analysis be integrated into designing and implementation of all SMS projects and programs to minimize overlaps and duplication of efforts among sanitation actors. It is further recommended that deliberate and sufficient budgetary resources be allocated to all key players to enable execution of their respective mandates. Similarly, our recommendation to practitioners is that stakeholder analysis be incorporated into project implementation frameworks to keep track of stakeholder movements and thereby align engagement strategies accordingly.

There is need for further research on how low power, high interest stakeholders can be empowered to enable them be active participants with homegrown solutions to the sanitation challenges they faced. This may involve interrogating the effectiveness of the current participatory governance models in Zambia to deliver SMS at household level.

## Author contributions

**Conceptualization:** Beatrice Chiwala.

**Data curation:** Beatrice Chiwala.

**Funding acquisition:** Beatrice Chiwala.

Investigation: Beatrice Chiwala.

Methodology: Beatrice Chiwala, Joseph Mumba Zulu.

Project administration: Mpundu Makasa, Joseph Mumba Zulu.

Supervision: Mpundu Makasa, Joseph Mumba Zulu.

Validation: Mpundu Makasa, Joseph Mumba Zulu.

Writing – original draft: Beatrice Chiwala.

Writing – review & editing: Beatrice Chiwala, Mpundu Makasa, Joseph Mumba Zulu.

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
