## [Decision Letter · Decision Letter 0]

28 Jan 2025

Dear Dr. Chiwala¹,

Thank you for submitting your manuscript to PLOS ONE. After careful consideration, we feel that it has merit but does not fully meet PLOS ONE’s publication criteria as it currently stands. Therefore, we invite you to submit a revised version of the manuscript that addresses the points raised during the review process.

We look forward to receiving your revised manuscript.

Kind regards,

D. Daniel, Ph.D.

Academic Editor

PLOS ONE

“This research article is supported through the Norwegian Programme for Capacity Development in Higher Education and Research for Development (NORHED–II/) and Strengthening Health Systems through Primary Care Leaders’ Education (PRICE) project scholarship BC received in partnership with the University of Zambia.”

Reviewers' comments:

Reviewer's Responses to Questions

**Comments to the Author**

1. Is the manuscript technically sound, and do the data support the conclusions?

Reviewer #1: Yes

Reviewer #2: Partly

2. Has the statistical analysis been performed appropriately and rigorously?

Reviewer #1: N/A

Reviewer #2: N/A

3. Have the authors made all data underlying the findings in their manuscript fully available?

Reviewer #1: Yes

Reviewer #2: Yes

4. Is the manuscript presented in an intelligible fashion and written in standard English?

Reviewer #1: Yes

Reviewer #2: Yes

Reviewer #1: 1. Abstract: It is stated in the introduction that ‘the objective of the study was to evaluate how stakeholders perceived their own power and interest in the context of providing SMS’. This objective is not clearly stated in the abstract.

2. Abstract, results section (use of plural or singular): The sentence ‘the main results suggested that stakeholders displayed interrelationships that was symbiotic’ to read the main results suggested that stakeholders displayed interrelationships that were symbiotic’.

3. Introduction, second paragraph: I suggest that the definition of access to SMS is made earlier in the introduction as it will enhance readers’ understanding and appreciation of the related concepts and issues presented before the definition is made.

4. Data collection and sampling strategy: It seems that some transect walks were longer than 15 minutes as interpreted from this sentence, ‘Before each walk, the team planned what to observe, estimated duration...’. If so, the authors should edit the statement, ‘from 06 AM to 07 AM for 15 minutes’ to ‘from 06 AM to 07 AM for about 15 minutes’.

5. Data analysis: The authors used several data collection approaches including online and phone-based approaches. It is not clear though how data collected from the different approaches were collated and analysed. It also not very clear how NVivo was used to analyse the data. They should elaborate. The data analysis process they present seems to be manual.

6. Data analysis: first sentence of the final paragraph in this section reads ‘The final stage of the analysis culminated in the production of this study report’. I suggest they edit to ‘this manuscript’ or ‘this paper’.

7. Results: One quotation on page 13 does not directly relate to the data it is supposedly reinforcing/ providing evidence for (TW1). (P17 KII) may also require relooking at.

8. Page 14: Last paragraph before discussion is about coordination and not about duo roles or shifting mandates. I suggest the authors move this paragraph and corresponding quotation to appropriate section of the manuscript.

9. Discussion: a couple of sentences start with or contain the phrase ‘we saw’. This suggests observations were carried out when in fact not or simply needs tight writing.

10. Discussion, page 14- last sentence to page 15 – first three sentences reference Chirgwin and Kennedy Walker as having ‘demonstrated that [the] powerful stakeholders with policy, regulatory and financial power were strategic in ensuring there were resources to implement SMS besides creating an enabling environment in terms of....’. First, the authors should delete ‘the’ as indicated in the last sentence above. Secondly, a reference like this would enhance the findings if more information is provided about the studies being referred to.

11. Discussion, page 15, third paragraph starting with ‘Nevertheless, relying on low power high interest stakeholders alone..’. I found this paragraph difficult to follow logically. I suggest the authors rewrite it.

Reviewer #2: Title: Power and interest levels in safely managed sanitation services in Zambia: a stakeholder mapping

Authors: Beatrice Chiwala; Mpundu Makasa; and Joseph Mumba Zulu

General comments:

Beatrice Chiwala et al. conducted the study that addresses an important and under-researched topic—stakeholder dynamics in the provision of safely managed sanitation services (SMS) in peri-urban areas of Lusaka, Zambia. The overall aim of this study is evaluating how stakeholders perceive their power and interest in providing SMS. Additionally, the authors analyzed the impact of stakeholders' quadrant positions (as per Mendelow's framework) on their capacity to contribute to SMS implementation. The authors also try to explore how shifting stakeholder roles and interdependencies influence sanitation outcomes. The study's use of Mendelow's Stakeholder Matrix is innovative and provides valuable insights into the interrelationships, power dynamics, and interests of various actors involved in SMS implementation. The findings have the potential to guide policy and practice in similar low- and middle-income settings.

Abstract:

The abstract is informative but lacks specificity about the findings' implications for policy or practice.

Introduction:

The introduction effectively establishes the significance of the problem and situates the study within global and regional contexts. However, it would benefit from a stronger articulation of the research gap and a clearer link between the study objectives and the theoretical framework.

Methods:

The narrative qualitative design is appropriate for the research objectives, and the combination of key informant interviews, focus group discussions, and transect walks provides a robust dataset. However, the manuscript would benefit from more detail on how themes were derived, including coding processes and triangulation strategies, to enhance the credibility of the findings.

Results:

The results section is detailed and presents a clear application of Mendelow's framework. However, the presentation would be improved by adding figures or tables to visually depict the stakeholder matrix and interrelationships.

The manuscript mentions various stakeholder categories (e.g., government institutions, NGOs, community members) and provides examples of their involvement in safely managed sanitation (SMS) services. However, the manuscript would benefit significantly from a clearer and more detailed description of each stakeholder's specific roles and responsibilities in SMS interventions. For example:

• Government Institutions: What specific policies, regulations, or enforcement mechanisms are implemented by ministries like the Ministry of Health or the Ministry of Water Development and Sanitation? How do these institutions support or hinder SMS efforts?

• NGOs and Community-Based Organizations: How do advocacy and awareness-raising activities translate into tangible SMS outcomes? Are there specific programs or models these organizations employ to address challenges in peri-urban areas?

• Beneficiaries: What behaviors or practices among the community impact SMS outcomes, and how can these stakeholders be more effectively empowered?

I suggest to introduce a section or table summarizing the roles, responsibilities, and contributions of each stakeholder type to enhance the readers' understanding of the dynamics in the study context and provide a clearer link between stakeholder actions and SMS outcomes.

Discussion:

The discussion highlights key findings but often reiterates results rather than critically interpreting them. It could be enhanced by drawing more explicit connections between the findings and broader literature, as well as by proposing actionable recommendations for policy and practice.

The use of the Mendelow Stakeholder Matrix is a novel and valuable approach. However, the manuscript could better articulate how the findings align with or validate the matrix's framework. For instance:

• Dynamic Shifts: The manuscript highlights that stakeholders often shift quadrants (e.g., local authorities performing dual roles). Explain how this aligns with the theoretical flexibility of the Mendelow Matrix and what implications these shifts have for stakeholder engagement strategies.

• Power-Interest Analysis: The manuscript describes quadrant categorizations, but it does not delve deeply into how the specific power-interest dynamics influence SMS outcomes. For example, how do stakeholders in the “low power, high interest” quadrant (e.g., beneficiaries) contribute to or limit program success? How can stakeholders with “high power, low interest” be better incentivized to engage more actively?

My suggestion is the authors could expand the discussion section to explicitly connect the matrix framework to the study's findings. For example, explain how Mendelow's theory helps interpret stakeholder behavior, prioritize engagement strategies, and address challenges such as overlapping mandates or siloed interventions.

Based on the stakeholder roles and Mendelow analysis, the authors may propose concrete strategies for improving stakeholder coordination, engagement, and resource allocation. For instance:

• How should government agencies address overlapping mandates?

• What mechanisms can enhance the participation of low-power stakeholders?

• How can NGOs and donors better align with national strategies to avoid silos?

Conclusion:

The conclusion effectively summarizes the findings but does not sufficiently emphasize the study’s implications or provide a clear roadmap for future research or interventions.

The manuscript concludes with general observations on stakeholder interdependence but lacks specific, actionable recommendations for practitioners and policymakers.

Recommendation:

The manuscript demonstrates strong potential but requires some revisions to improve its clarity, depth of analysis, and presentation. Suggested revisions include:

• Strengthening the introduction by explicitly stating the research gap.

• Expanding on methods for data analysis and coding validation.

• Enhancing the discussion with critical analysis and actionable recommendations.

• Improving visual presentation of the stakeholder matrix.

Language:

Review the manuscript for minor grammatical errors and ensure adherence to standard academic English.

Abstract

Line 11: The phrasing "stakeholder influence access to SMS" is unclear and grammatically incorrect. This could be revised to: "Variations in power and interest among stakeholders significantly influence access to SMS."

Lines 21 – 22: "Similarly, stakeholders within and outside the quadrant were not homogenous". This statement is vague and lacks context for what "within and outside the quadrant" refers to. To provide clarity, I would suggest to write: "Stakeholders categorized into the Mendelow quadrants displayed varying levels of homogeneity in power and interest."

Introduction

Lines 36 – 38: The sentence is too long and awkwardly structured. This could be break into two sentences, for example: "In sub-Saharan Africa (SSA), a majority of those affected live in peri-urban and rural areas. They predominantly rely on non-sewered, unimproved sanitation facilities."

Lines 51 – 53: The sentence is overly detailed for an introduction. This could be simplified to: "Access to SMS entails improved sanitation facilities with safely managed waste disposal, either in situ or off-site."

Methods

Lines 80 – 82: The sentence is verbose and unclear. This could be simplified to: "The narrative research design enabled the collection of data on stakeholders' roles in providing SMS at the household level."

Results

Lines 188 – 189: Repetition of "stakeholders" and "levels". Could be revised to: "Using respondents' input, we identified stakeholders and evaluated their power and interest levels through Mendelow’s matrix."

Line 372: "The results showed how the stakeholder are interrelated in an effort to provide SMS". Grammatical error ("stakeholder" should be plural).

Discussion

Lines 554 – 556: The sentence is long and cluttered. This could be simplified to: "During emergencies, stakeholders with high power stepped in to provide resources, including financial, material, and technical support, particularly for infrastructure development."

Conclusion

Lines 611: Repetitive phrasing ("levels of power and interest"). Suggestion for revision: "Achieving SMS requires collective efforts from all stakeholders, regardless of their power and interest levels."

**Do you want your identity to be public for this peer review?** For information about this choice, including consent withdrawal, please see our Privacy Policy

Reviewer #1: No

Reviewer #2: No

---

## [Author Response · Author response to Decision Letter 1]

11 Jul 2025

Response: Thanks. We have revised all the titles to required font size in entire manuscript and included author affiliations as advised

“This research article is supported through the Norwegian Programme for Capacity Development in Higher Education and Research for Development (NORHED–II/) and Strengthening Health Systems through Primary Care Leaders’ Education (PRICE) project scholarship BC received in partnership with the University of Zambia.”

Response: Thanks. The statement proposed is correct as stated. "The funders had no role in study design, data collection and analysis, decision to publish, or preparation of the manuscript." Kindly proceed to change online accordingly.

Response: Thanks. We have included captions for our supporting information at the end of the manuscript as guided.

Reviewers' comments:

Reviewer's Responses to Questions

Comments to the Author

1. Is the manuscript technically sound, and do the data support the conclusions?

Reviewer #1: Yes

Reviewer #2: Partly

Response: Thank you for the feedback.

2. Has the statistical analysis been performed appropriately and rigorously?

Reviewer #1: N/A

Reviewer #2: N/A

Response: Thank you for the feedback

3. Have the authors made all data underlying the findings in their manuscript fully available?

Reviewer #1: Yes

Reviewer #2: Yes

Response: Thank you for the feedback

4. Is the manuscript presented in an intelligible fashion and written in standard English?

Reviewer #1: Yes

Reviewer #2: Yes

Response: Thank you for the positive feedback

5. Review Comments to the Author

Reviewer #1: 1. Abstract: It is stated in the introduction that ‘the objective of the study was to evaluate how stakeholders perceived their own power and interest in the context of providing SMS’. This objective is not clearly stated in the abstract.

Response: Thanks. We have since revised the study objective and included it in the abstract in line 14 to 17 to read as:

The study aimed to explore and analyze how stakeholders perceived their power and interest in the context of providing SMS. The study applied the Mendelow Stakeholder Matrix to identify, characterize and analyze the actors involved in the provision of SMS in peri-urban areas in Lusaka, Zambia.

2. Abstract, results section (use of plural or singular): The sentence ‘the main results suggested that stakeholders displayed interrelationships that was symbiotic’ to read the main results suggested that stakeholders displayed interrelationships that were symbiotic’.

Response: Thanks. We have revised the sentence accordingly in line 24 and 25 to read:

The main results suggested that stakeholders displayed interrelationships that were symbiotic as they depended on each other to deliver their mandates

3. Introduction, second paragraph: I suggest that the definition of access to SMS is made earlier in the introduction as it will enhance readers’ understanding and appreciation of the related concepts and issues presented before the definition is made.

Response: Thanks. We have since moved the definition of access to SMS to line 44-46 to read:

Access to SMS entails improved sanitation facilities with safely managed waste disposal, either in situ or off-site. (2,3)

4. Data collection and sampling strategy: It seems that some transect walks were longer than 15 minutes as interpreted from this sentence, ‘Before each walk, the team planned what to observe, estimated duration...’. If so, the authors should edit the statement, ‘from 06 AM to 07 AM for 15 minutes’ to ‘from 06 AM to 07 AM for about 15 minutes’.

Response: Thanks. We have since revised to include ‘about 15 minutes in line 146-147 as guided to now read:

The walks were conducted early in the morning, from 06 AM to 07 AM for about 15 minutes, allowing the team to witness how households managed sanitation issues.

5. Data analysis: The authors used several data collection approaches including online and phone-based approaches. It is not clear though how data collected from the different approaches were collated and analysed. It also not very clear how NVivo was used to analyse the data. They should elaborate. The data analysis process they present seems to be manual.

Response: Thanks. We have revised the section to include how data collected from different approaches was collated and analysed in line 174-191 to now read:

All audio recorded transect walks, focus group discussions and key informant interviews were transcribed by an independent transcriber. BC sampled the transcripts against the audio files to validate their accuracy. All transcripts were cleaned and imported into NVivo release 14.23.2 (46) for coding and analysis. NVivo is a qualitative tool used for managing, organizing and analyzing data (50–52). A reflective thematic analysis inspired by Braun and Clark (53) was applied to analyze all transcribed data using inductive and deductive coding approaches (54–57). Inductive and deductive processes were applied during the coding process to enhance the rigor of created codes (55,56) Inductive approach was utilized to identify patterns on stakeholders’ perceptions, views and ideas regarding their power and interest in the provision of safely managed sanitation services. This process involved reading and re-reading the transcripts to identify significant information that would guide the mapping of stakeholders according to their power and interest. After identifying a series of codes, the research team explored connections between the identified codes and categorized stakeholders based on their similarities. Through the deductive process, the final codes were subsequently mapped onto the four quadrants of the Mendelow stakeholder matrix reflecting the sanitation actors’ levels of power and interest. The subsequent stages involved reviewing and refining the themes to align with the study's objectives through a consultative process. Figure 2 show the final Mendelow stakeholder matrix.

6. Data analysis: first sentence of the final paragraph in this section reads ‘The final stage of the analysis culminated in the production of this study report’. I suggest they edit to ‘this manuscript’ or ‘this paper’.

Response: Thanks. The sentence as since been revised in line 192 to read as:

The final stage of the analysis culminated in the production of this manuscript.

7. Results: One quotation on page 13 does not directly relate to the data it is supposedly reinforcing/ providing evidence for (TW1). (P17 KII) may also require relooking at.

Response: Thanks. We have since replaced the quotations in lines 474 -478 for (TW1) and 509 -514 for (P17 KII) to read respectively:

“What makes people not to have toilets it's because of lack of money which makes it very difficult for them to construct toilets and also you talked about the law which talks about having a toilet at every household it's true but to have a standard toilet you require toilet with soak away whereby when they are full they can be emptied and most people did not know about those things it’s just now that people have become aware about them.” (TW1).

“So, there are all these issues at institution or WASH as a mandate that remains with respective institutions like Ministry of Health, Ministry of Education, they also had institutional WASH mandate was reporting to who, who is tracking what is happening and this is why we are really unable to articulate the impacts that is being made towards achieving SDG six because everyone has something that they are doing and there's no central coordination on in terms of the data that is going”. (P17, KII)

8. Page 14: Last paragraph before discussion is about coordination and not about duo roles or shifting mandates. I suggest the authors move this paragraph and corresponding quotation to appropriate section of the manuscript.

Response: Thanks. We have since included a sub-title to accommodate limited coordination challenges in lines 515-523 to now read:

Limited collaboration and coordination among stakeholders

Results also revealed coordination challenges among stakeholders as some preferred to implement sanitation interventions in isolation. This contributed to duplication of efforts and negatively affected coordination mechanisms and resource mobilization in the sector.

“I think the biggest challenge is working in silos. When we have a lot of partners, they always want to be visible, being seen as the ones spearheading everything, yes we may come, we may come together in the WASH committee”…….. But in terms of implementing, I think most of these institutions would want to be seen as the ones championing, improvement of sanitation in a specific area. So there's need to harmonize whatever we're doing and to avoid working in silos”. (P33, KII)

9. Discussion: a couple of sentences start with or contain the phrase ‘we saw’. This suggests observations were carried out when in fact not or simply needs tight writing.

Response: Thanks. We have removed ‘we saw’ phrases wherever it appeared as reflected in lines 537, 574 -576 and 589-590 respectively to read as:

a) Stakeholders like the NGOs got involved in resource mobilization besides their primary function of advocacy and awareness raising.

b) It was also during emergencies in our findings that stakeholders such as the Local Authority shifted quadrants and performed dual roles as they interpreted and enforced the Public Health Act.

c) On the other hand, findings revealed that some community members were unconcerned about the poor sanitation status around them

10. Discussion, page 14- last sentence to page 15 – first three sentences reference Chirgwin and Kennedy Walker as having ‘demonstrated that [the] powerful stakeholders with policy, regulatory and financial power were strategic in ensuring there were resources to implement SMS besides creating an enabling environment in terms of....’. First, the authors should delete ‘the’ as indicated in the last sentence above. Secondly, a reference like this would enhance the findings if more information is provided about the studies being referred to.

Response: Thanks. ‘the’ has since been deleted and reference has since been inserted from lines 540-543 to now read:

In similar studies, Chirgwin et al and Kennedy-Walker with his co–authors demonstrated that powerful stakeholders with policy, regulatory and financial power were strategic in ensuring there were resources to implement SMS besides creating an enabling environment in terms of legal, policy and institutional framework (8,62).

11. Discussion, page 15, third paragraph starting with ‘Nevertheless, relying on low power high interest stakeholders alone..’. I found this paragraph difficult to follow logically. I suggest the authors rewrite it.

Response: Thanks. We have since revised the paragraph from 562-570 to read as

Nevertheless, relying on low power/high interest stakeholders alone without those with high power/ high interest might not be enough if SMS was to be attained. Studies have revealed that high poverty levels among the low power/high interest stakeholders, coupled with lack of land tenure for most residents in peri–urban areas compound the problem as revealed in this study and other studies (84–86). As the results revealed, there were households that had no sanitation facilities for years despite the various interventions in the study target areas. This was where enforcement and regulation played a critical role to compel households to construct sanitation facilities (84,87). Additionally as our results have shown, NGOs and community–level stakeholders also come in to raise awareness and advocate for improved sanitation at household and community level (88–90).

Reviewer #2: Title: Power and interest levels in safely managed sanitation services in Zambia: a stakeholder mapping

Authors: Beatrice Chiwala; Mpundu Makasa; and Joseph Mumba Zulu

General comments:

Beatrice Chiwala et al. conducted the study that addresses an important and under-researched topic—stakeholder dynamics in the provision of safely managed sanitation services (SMS) in peri-urban areas of Lusaka, Zambia. The overall aim of this study is evaluating how stakeholders perceive their power and interest in providing SMS. Additionally, the authors analyzed the impact of stakeholders' quadrant positions (as per Mendelow's framework) on their capacity to contribute to SMS implementation. The authors also try to explore how shifting stakeholder roles and interdependencies influence sanitation outcomes. The study's use of Mendelow's Stakeholder Matrix is innovative and provides valuable insights into the interrelationships, power dynamics, and interests of various actors involved in SMS implementation. The findings have the potential to guide policy and practice in similar low- and middle-income settings.

Response: Thank you for the feedback.

Abstract:

The abstract is informative but lacks specificity about the findings' implications for policy or practice.

Response: Thanks. We have since included implications for policy and practice as guided in line 32-36 to read as:

Policy implications, especially to Government, may mean allocation of adequate resources to key players to enable them deliver on their respective mandates. Similarly, implications to practitioners might be the need to periodically review stakeholders and forge alliances coupled with conducting multi-sectoral meeting ai

---

## [Decision Letter · Decision Letter 1]

6 Aug 2025

Dear Dr. Chiwala¹,

Thank you for submitting your manuscript to PLOS ONE. After careful consideration, we feel that it has merit but does not fully meet PLOS ONE’s publication criteria as it currently stands. Therefore, we invite you to submit a revised version of the manuscript that addresses the points raised during the review process.

We look forward to receiving your revised manuscript.

Kind regards,

D. Daniel, Ph.D.

Academic Editor

PLOS ONE

Journal Requirements:

Reviewers' comments:

Reviewer's Responses to Questions

**Comments to the Author**

Reviewer #2: All comments have been addressed

2. Is the manuscript technically sound, and do the data support the conclusions?

Reviewer #2: Yes

3. Has the statistical analysis been performed appropriately and rigorously?

Reviewer #2: N/A

4. Have the authors made all data underlying the findings in their manuscript fully available?

Reviewer #2: Yes

5. Is the manuscript presented in an intelligible fashion and written in standard English?

Reviewer #2: Yes

Reviewer #2: The revised manuscript demonstrates a strong effort to address and respond to my previous comments. Key structural, conceptual, and linguistic revisions were incorporated in several sections. However, some minor adjustments would improve the manuscript. For example: in results section, including a diagram of stakeholder categories by quadrant would greatly enhance the reader’s comprehension.

**Do you want your identity to be public for this peer review?** For information about this choice, including consent withdrawal, please see our Privacy Policy

Reviewer #2: No

---

## [Author Response · Author response to Decision Letter 2]

12 Sep 2025

Journal Requirements:

Response: Thank you for the suggestion. We have since reviewed the reference list to ensure that it is complete and correct. As part of the revisions, we have added/updated the DOI and URL where applicable/available to improve accessibility of the cited sources. We have also reviewed and evaluated the cited articles and confirmed that they are still relevant to the study.

Reviewers' comments:

Reviewer's Responses to Questions

Comments to the Author

1. If the authors have adequately addressed your comments raised in a previous round of review and you feel that this manuscript is now acceptable for publication, you may indicate that here to bypass the “Comments to the Author” section, enter your conflict of interest statement in the “Confidential to Editor” section, and submit your "Accept" recommendation.

Reviewer #2: All comments have been addressed

Response: Thank you for the positive feedback.

2. Is the manuscript technically sound, and do the data support the conclusions?

Reviewer #2: Yes

Response: Thank you for the positive feedback.

3. Has the statistical analysis been performed appropriately and rigorously?

Reviewer #2: N/A

Response: Thank you for the positive feedback.

4. Have the authors made all data underlying the findings in their manuscript fully available?

Reviewer #2: Yes

Response: Thank you for the positive feedback.

5. Is the manuscript presented in an intelligible fashion and written in standard English?

Reviewer #2: Yes

Response: Thank you for the positive feedback.

6. Review Comments to the Author

Reviewer #2: The revised manuscript demonstrates a strong effort to address and respond to my previous comments. Key structural, conceptual, and linguistic revisions were incorporated in several sections. However, some minor adjustments would improve the manuscript. For example: in results section, including a diagram of stakeholder categories by quadrant would greatly enhance the reader’s comprehension.

Response: Thank you for the valuable feedback. We have inserted four tables highlighting the grouped sanitation stakeholders per quadrant as illustrated in lines 227 to 233.

---

## [Decision Letter · Decision Letter 2]

8 Oct 2025

Power and interest levels in safely managed sanitation services in Zambia: a stakeholder mapping

PONE-D-24-50711R2

Dear Dr. Chiwala¹,

We’re pleased to inform you that your manuscript has been judged scientifically suitable for publication and will be formally accepted for publication once it meets all outstanding technical requirements.

Kind regards,

D. Daniel, Ph.D.

Academic Editor

PLOS ONE

Additional Editor Comments (optional):

Reviewers' comments:

Reviewer's Responses to Questions

**Comments to the Author**

Reviewer #2: All comments have been addressed

2. Is the manuscript technically sound, and do the data support the conclusions?

Reviewer #2: Yes

3. Has the statistical analysis been performed appropriately and rigorously?

Reviewer #2: N/A

4. Have the authors made all data underlying the findings in their manuscript fully available?

Reviewer #2: Yes

5. Is the manuscript presented in an intelligible fashion and written in standard English?

Reviewer #2: Yes

Reviewer #2: Appreciation to all authors for the strong and good works in responding all comments and suggestions.

**Do you want your identity to be public for this peer review?** For information about this choice, including consent withdrawal, please see our Privacy Policy

Reviewer #2: No

---

## [Editor Report · Acceptance letter]

PONE-D-24-50711R2

PLOS ONE

Dear Dr. Chiwala¹,

I'm pleased to inform you that your manuscript has been deemed suitable for publication in PLOS ONE. Congratulations! Your manuscript is now being handed over to our production team.

Kind regards,

on behalf of

Dr. D. Daniel

Academic Editor

PLOS ONE